# Elicitors and Pre-Fermentative Cold Maceration: Effects on Polyphenol Concentration in Monastrell Grapes and Wines

**DOI:** 10.3390/biom9110671

**Published:** 2019-10-30

**Authors:** Diego F. Paladines-Quezada, Juan D. Moreno-Olivares, José I. Fernández-Fernández, Juan A. Bleda-Sánchez, Alejandro Martínez-Moreno, Rocío Gil-Muñoz

**Affiliations:** 1Instituto Murciano de Investigación y Desarrollo Agrario y Alimentario (IMIDA), Ctra. La Alberca s/n, 30150 Murcia, Spain; diegopaladines@hotmail.com (D.F.P.-Q.); srjuanda@hotmail.es (J.D.M.-O.); josei.fernandez@carm.es (J.I.F.-F.); juanantonio.bleda@carm.es (J.A.B.-S.); 2Centro de Edafología y Biología Aplicada del Segura (CEBAS-CSIC), Campus Universitario de Espinardo, 30100 Murcia, Spain; amartinez@cebas.csic.es

**Keywords:** phenolic compounds, methyl jasmonate, benzothiadiazole, flavonols

## Abstract

*Vitis vinifera* L cv Monastrell is the main red grape variety grown for vinification in the Denomination of Origin Jumilla (southeast Spain). Different strategies are still being tested to optimize available resources both in terms of the environment and to achieve wines with better organoleptic and functional characteristics. The objective of this work was to combine two strategies: the application of methyl jasmonate (MeJ), benzothiadiazole (BTH), and methyl jasmonate + benzothiadiazole (MeJ + BTH) elicitors to Monastrell leaves, and pre-fermentative cold maceration. During two seasons, the experiment was carried out to improve the extraction of the phenolic compounds, whose levels may have increased following the application of elicitors in the field, and to assess the effect of both strategies on the wine quality. Discriminant analysis revealed that independently of the meteorological conditions during both years, the pre-harvest response to the application of elicitors MeJ, BTH, and MeJ + BTH, induced increases in total anthocyanin concentration of the treated grapes, allowing the distinction of the treatments. This analysis also allowed the distinction between the types of maceration used, showing greater extraction of phenolic compounds by the application of low temperature, giving wines with a higher index of total phenols, a greater intensity of color, and a lower luminosity.

## 1. Introduction

*Vitis vinifera* L. cv Monastrell is the fourth most cultivated red wine grape variety in Spain, where Murcia and Alicante are its main production areas [1]. This variety represents over 80% of the vineyards in the Denomination of Origin Jumilla [2], and is very well adapted to the soil and weather conditions of the area (southeast Spain). It has a high content of phenolic compounds, which are difficult to fully extract during vinification [3]. Analysis of the composition and morphology of the Monastrell skin cell walls showed that its skin is thicker than Syrah and Cabernet Sauvignon [4], which could explain the difficulty of extraction of the phenolic compounds during maceration. This is why, in recent years, more forceful oenological techniques have been studied to help modulate the content of phenolic compounds in Monastrell wines, such as different maceration times [5], the use of low temperature pre-fermentative maceration, must freezing with dry ice, and the use of a maceration enzyme [6]. Among the oenological techniques used in Monastrell, pre-fermentative cold maceration has proved to be an effective method extracting anthocyanins, volatile compounds, and proanthocyanidins more efficiently [6,7].

Different strategies are still being tested in this region in an attempt to optimize the use of available resources with respect to the environment and to improve the polyphenolic quality of the main vine varieties cultivated in order to obtain wines with better organoleptic and functional characteristics since the organoleptic properties of red wines depend on the phenolic compounds [7], and these compounds in turn have shown to have beneficial properties for health, including antioxidant, anticarcinogenic, and antispasmodic properties, enhancement or activation of bile secretion, and antibacterial and antihistaminic agents [8], demonstrating their functional characteristic.

Among these strategies are clonal and sanitary selection programs of the Monastrell [1], deficit irrigation [9], and the foliar application of different elicitors [10,11,12,13,14]. Elicitors were first used to increase plant resistance to pathogens, although it was later found that the mechanism involved increased polyphenol levels. This strategy consists of stimulating and/or potentiating the grapevine defense response by applying exogenous molecules often originating from microbes or plants [11]. Application of elicitors in a concentrated form in a week during the veraison to increase the phenolic compounds has been described [15], that is the critical period during which the developmental changes of polyphenols assume an important role [16]. Some elicitors, such as methyl jasmonate (MeJ) and benzothiadiazole (BTH), have been seen to show a positive effect on increasing the phenolic composition of Monastrell and other varieties [10,12,13,17].

However, the increases achieved in the fresh skins of Monastrell have not been reflected in any great extent in the wines produced, as observed in a previous study [18]. For this reason, the objective of this work was to combine two strategies: the foliar application of two elicitors (MeJ, BTH and MeJ + BTH) in the Monastrell variety, and pre-fermentative cold maceration, in order to guarantee a better extraction of phenolic compounds and obtain higher quality wines.

## 2. Materials and Methods 

### 2.1. Experimental design

The experiment was carried out over two consecutive years (2016–2017) on the Monastrell variety located in Jumilla, Murcia (southeastern Spain). The study was performed on 14 year old *Vitis vinifera* (syn. Mourvedre) red wine grapevines grafted onto 1103-Paulsen (clone 249) rootstock and trained to a three-vine vertical trellis system. Vine rows were arranged from N–NW to S–SE with the between-row and within-row spacing of 3 x 1.25 m (x: 636.099; y: 4.249.299).

All treatments were applied to three replicates and were arranged in a complete randomized block design, with 10 vines for each replication. The protocol used to apply the different treatments, as well as the doses used, have been described previously [10]. Plants were sprayed with a water suspension of two elicitors: MeJ (methyl jasmonate; Sigma Aldrich, St. Louis, USA), at a concentration of 10 mM; BTH (benzo-(1,2,3)-thiadiazole-7-carbothioic acid S-methyl ester; Sigma Aldrich, St. Louis, USA) at a concentration of 0.3 mM, and a mixture of both (methyl jasmonate at a concentration of 10 mM and BTH at a concentration of 0.3 mM). To carry out the treatments, aqueous solutions (200 mL per plant) were prepared with Tween 80 (Sigma Aldrich, St. Louis, USA) as the wetting agent (0.1% *v/v*). Control plants were sprayed with aqueous solution of Tween 80 alone. The treatments were carried out twice, at veraison and 1 week later. When grapes reached technological maturity (maximum sugar/acidity ratio, determined by the control grapes), they were harvested and transported in boxes to the winery for physicochemical analysis and vinification.

### 2.2. Physicochemical analysis in grapes at harvest

Total soluble solids (°Brix) were measured using an Abbé-type refractometer (Atago RX-5000). Titratable acidity was measured using an automatic titrator (Metrohm, Herisau, Switzerland) with 0.1 N NaOH. Tartaric and malic acids were measured using enzymatic kits from Boehringer 146 Mannheim GmbH (Mannhein, Germany). The methodology used to carry out these analyses is described in OIV (Organisation Internationale de la Vigne et du Vin) (2018) [19].

### 2.3. Vinifications

All vinifications were made in triplicate in 50-L stainless steel tanks using 50 kg of grapes, which were destemmed, crushed, and sulfited (8 g SO_2_/100 kg). Total acidity was corrected to 5.5 g/L with tartaric acid, and selected yeasts were added (Uvaferm VRB, Lallemand, 25 g/hL). Two types of vinification were carried out: a pre-fermentative cold maceration (PFCM) followed by traditional maceration (TM), and another using only a traditional maceration.

*Pre-fermentative cold maceration (PFCM):* Tanks containing crushed grapes were placed in a cold room at 4 °C for 10 days, after which the tanks were returned to the winery to continue the traditional maceration.

*Traditional maceration (TM):* All vinifications were conducted at 25±1 °C. The fermentative pomace contact period was 10 days. Throughout the fermentation pomace contact period, the cap was punched down twice a day, and the temperature and must density were recorded. At the end of this period, wines were pressed at 1.5 bars in a 75 L tank membrane press. Free-run and press wines were combined and stored at room temperature. The analyses were carried out at the end of alcoholic fermentation (AF) in triplicate.

### 2.4. Analysis of anthocyanins and flavonols in grapes and wines

Grapes were peeled with a scalpel, and the skins were stored at –20 °C until analysis. Samples (2 g) were immersed in methanol (40 mL) in hermetically closed tubes and placed on a stirring plate at 150 rpm and 25 °C. After 4 h, the methanolic extracts were filtered through the 0.45 μm nylon filters and directly analyzed by HPLC, according to Gil-Muñoz et al. [10]. The HPLC analyses were performed on a Waters 2690 liquid chromatograph (Waters, Milford, PA, USA), equipped with a Waters 996 diode array detector and a Licrochart RP-18 column (Merck, Darmstadt, Germany), 25 · 0.4 cm, 5 lm particle size, using as solvents water plus 5% formic acid (solvent A) and HPLC grade methanol (solvent B) at a flow rate of 1 mL min^−1^. Elution was performed with a gradient starting with 2% B to reach 32% B in 40 min, isocratic for 15 min, 50% B at 70 min, 60% B at 75 min and then isocratic for 5 min. Chromatograms were recorded at 280 nm (phenolic acids and flavan-3-ols), 320 nm (hydroxycinnamic acids and derivatives), 360 nm (flavonols), and 520 nm (anthocyanins). Anthocyanins were quantified at 520 nm as malvidin-3-glucoside, using malvidin-3-glucoside chloride as external standard (Extrasynthèse, Genay, France). Phenolic acids and flavan-3-ols were quantified at 280 nm, and flavonols at 360 nm using the pure compounds as external standards. For the quantification of hydroxycinnamic derivatives, caftaric acid was quantified as caffeic acid and coutaric acid as coumaric acid.

### 2.5. Spectrophotometric parameters in wine

Wine was previously centrifuged and color intensity (CI), CIELab parameters, total phenols (TP) and total anthocyanins (TA) were analyzed by a Shimadzu UV–vis spectrophotometer, model 1600PC (Shimadzu, Duisburg, Germany), as described in Paladines-Quezada et al. [18]. CI was calculated as the sum of absorbance at 620 (blue component), 520 (red component), and 420 nm (yellow component) in undiluted wine [20]. The CIELab parameter L* (lightness) was determined by measuring the transmittance of the wine every 10 nm from 380 to 770 nm, using the D65 illuminant and a 10° observer angle. TP was calculated by measuring wine absorbance at 280 nm, according to Ribéreau-Gayon, Pontallier, and Glories [21] and TA by the method proposed by Ho, Silva, and Hogg [22].

### 2.6. Statistical analysis

Significant differences among wines and grapes and for each variable were assessed by analysis of variance (ANOVA) using the Statgraphics 5.0 Plus package (Statpoint Technologies, Inc., Warrenton, VA, USA). The Duncan test was used to separate the means (*p* < 0.05) when the ANOVA test was significant; and a multivariate discriminant analysis was applied to identify the most discriminant variables.

## 3. Results and Discussion

### 3.1. Grapes

#### 3.1.1. HPLC analysis of Anthocyanins

The anthocyanins in grapes at harvest were quantified by HPLC (Table 1). The results for two consecutive campaigns involving treatment with MeJ, BTH and a mixture of both are presented. As can be seen, there was a clear difference between both years studied, 2016 and 2017. Total anthocyanins in 2016 had almost twice the concentration of those from 2017 (independently of the treatment used), a response that could be explained by the different climatic conditions during these two years (Figure 1A–C), especially in July and August. Likewise, the period between the beginning of the veraison and the technological maturation was affected. This period was 34 days in 2016 and 44 days in 2017.

The anthocyanin profile detected in the Monastrell variety coincided with that detected by Apolinar-Valiente et al. [23], Gil-Muñoz et al. [2], and Hernández-Jiménez et al. [24]. In agreement with these authors, Monastrell contained dihydroxylated anthocyanins (DiOH), cyanidin and peonidin-3-O-glucoside; trihydroxylated (TriOH) anthocyanins, delphinidin, petunidin and malvidin-3-o-glucoside; and acylated anthocyanins, acetates, coumarates andmalvidin-3-caffeate glucoside. The highest percentage corresponded to the anthocyanins monoglucosides, and malvidin-3-O-glucoside was the most abundant (Table 1), as is usual in young red wines [25,26].

During 2016, the treatments with the elicitors used (MeJ, BTH and MeJ + BTH) stimulated the polyphenol biosynthetic route, increasing the concentration of almost all the anthocyanins present in Monastrell (Table 1), such as DiOH, TriOH, acetates, and coumarates. The combination of MeJ + BTH, while increasing the anthocyanin concentrations compared with the control, did not show a synergic effect. Ruiz-García et al. [13] also reported that the joint application of MeJ + BTH on Monastrell grapes did not improve the concentration of anthocyanins compared with the separate application of the elicitors.

In 2017, the rainfall between July and August (Figure 1A) was greater than in the previous year (2.6 mm in 2016 and 54.2 mm in 2017), at a time when there is great physiological activity due to the growth and maturation of the berries. During this period, grapes accumulate a significant amount of water, which increases berry weight. This higher percentage of precipitation may have led to a higher absorption of humidity, accelerating berry growth through cell expansion [27]. Table 2 shows that the berries size during 2016 were smaller and had a higher concentration in °Brix than those of 2017. The results of the weight of 100 berries and °Brix of 2016 were presented in Paladines-Quezada et al. [18]. Reyero et al. [28] also observed that Monastrell grapes reached a greater size with increased rainfall, since the variety has a rapid response to water availability. This also results in a dilution of the phenolic compounds, which are mainly located in the skins [29]; since anthocyanins appear and accumulate from the time of veraison and during maturation [30].

To the greater precipitation of 2017 must be added a greater number of days with maximum temperatures above 30 °C (33 days; Figure 1C) compared to 2016 (22 days; Figure 1B), from the beginning of veraison until harvesting. Fernández-López et al. [31] showed that pigmentation in the Monastrell grape develops rapidly and, in the second week after veraison, the grape color is practically dark blue; although it has to be borne in mind that the accumulation and distribution of anthocyanins in the skin may vary from year to year depending on climatic conditions [32]. It has also been described that temperatures above 30 °C negatively influence sugar accumulation and anthocyanin levels [33], either by degradation or the inhibition of synthesis, or both [34,35]; and that interannual differences in a variety may be greater than intervarietal differences in the same year [36,37].

However, during 2017, the grapes treated with MeJ + BTH were the only ones that presented a higher concentration of anthocyanins/gram of skin with respect to the control grapes (Table 1), especially in anthocyanins glucosides such as delphinidin, cyanidin, petunidin, and peonidin, which was also confirmed from the total of non-acylated anthocyanins and DiOH. This time, the combination of MeJ + BTH elicitors may have shown a synergetic effect in stimulating the synthesis of anthocyanins mentioned above. By contrast, the grapes treated with MeJ registered a lower concentration of anthocyanins than control grapes, while the treatment with BTH did not produce significant differences compared to the control. The lower concentration of anthocyanins during 2017 could also be due to early technological maturation (maximum sugar/acidity ratio) as a result of high temperatures, bringing forward the beginning of veraison (approximately 15 days earlier than in 2016), and pushing back maturation of the phenolic compounds, thus avoiding an optimal maturation. In contrast, Pérez-Magariño et al. [38] and Sims and Bates [39] observed that wines from grapes harvested about two weeks after reaching technological maturity had higher color intensity and higher content of anthocyanins and total polyphenols than those from grapes at that point of maturity.

#### 3.1.2. HPLC analysis of Flavonols

An analysis of flavonols present in grapes was carried out (Table 3), since they are closely related to anthocyanins in the biosynthetic path. Flavonols are responsible for the yellow color of white wines, and, although they are found in lower proportions in red varieties, their presence is important since they form very stable copigmentation complexes with anthocyanins [40].

Independently of the treatments carried out, the results obtained in 2017 showed considerable increases in the concentration of isorhamnetin, as well as an important decrease in quercetin. As was mentioned above, the high temperatures that the grapes were exposed to this year (Figure 1C), together with a greater number of days between veraison and harvest, may have caused flavonols to increase, and the degradation or inhibition of quercetin, compared with the values recorded in 2016. Some authors point out that the biosynthesis of flavonols in grapes depends, to a greater extent than other phenolic compounds, on solar exposure, which favors their accumulation in grapes by increasing the hours of solar radiation [41,42,43]. Studies have been carried out on the effect of insolation on the phenolic composition of Syrah [44] and Merlot grapes [34], in which the authors observed that the concentration of flavonols, especially quercetin-3-glucoside, was favored in grapes with greater exposure to solar radiation. However, an excess of UV-B radiation can negatively affect the content of glucosylated flavonols [45].

The treatments carried out in the field during 2016 did not influence the concentration of any flavonol with respect to the values of control grapes. However, in 2017, treatments with MeJ and MeJ + BTH halved the concentrations of myricetin, and the concentration of quercetin in grapes treated with MeJ + BTH was also substantially reduced, so that these grapes were the only ones that presented a lower concentration of total flavonols compared to the control. However in the same year, the treatments with MeJ, BTH and MeJ + BTH produced a substantial increase in the concentration of kampferol with respect to the control grapes.

### 3.2. Wines

#### 3.2.1. Chromatic characteristics (spectrophotometric measurements)

The results for the 2016 vinification with traditional maceration (Figure 2) were presented in Paladines-Quezada et al. [18]. In an analysis of the chromatic characteristics of the wines (Figure 2 and Figure 3), the difference between the two years (independently of the treatments carried out in the field or the oenological technique used) were seen to be substantial. The values obtained in 2016 were almost twice higher in some cases than in 2017, which coincides with the results for the grapes (Table 1).

Thus, in 2016, wines made from grapes treated with MeJ and BTH using traditional maceration (TM) showed higher concentrations of total anthocyanins (Figure 2A), resulting in wines with a higher color intensity (Figure 2C). On the other hand, wines made with PFCM + TM showed much higher levels than those made with TM. The application of cold helped to extract a higher percentage of total anthocyanins, even in grapes treated with MeJ + BTH, which, in the case of TM, did not differ from control levels. The application of the PFCM + TM technique allowed the phenolic compounds to be extracted to a greater extent in those grapes that increased their phenolic composition through the use of elicitors, giving wines with a higher index of total phenols, a greater intensity of color and a lower luminosity. The priority diffusion of anthocyanins during the cold pre-fermentative phase may explain the increase in color normally observed in the wines so obtained [7]. These results show that anthocyanins from Monastrell skins are more difficult to extract, as other authors also indicated [3]; and may benefit from an extended cold maceration.

In the 2017 season, only wines made with grapes treated with BTH and PFCM + TM showed increases in the concentration of total anthocyanins compared to the control (Figure 3A), providing wines with greater color intensity and lower luminosity (Figure 3C,D). Ruiz-García et al. [12] also demonstrated that the wines obtained from Monastrell grapes treated with BTH showed higher color intensity and total phenolic content than the wines made from control grapes; these authors also indicated that this treatment could be of interest for obtaining wines with a deep and stable color and a potentially higher market value. Likewise, several studies have demonstrated the positive effect of applying low temperatures on the extraction of phenolic compounds [6,46].

#### 3.2.2. HPLC analysis of Anthocyanins

The wine anthocyanin concentration obtained by HPLC (Table 4 and Table 5) showed notable differences compared to the anthocyanin concentration analyzed in berries (Table 1). Revilla et al. [47] observed that this change seems to take place during fermentation, and that the differences found between grapes and wines may be due to differences in the extraction ratio of anthocyanins, degradation, and polymerization reactions and the different adsorption capacity in the yeast cell walls.

The wines elaborated using grapes treated with MeJ and TM during 2016 (Table 4) showed a higher concentration of total anthocyanins, the TriOH anthocyanins (delphinidin, petunidin, and malvidin-3-O-glucosides) were the only ones with a concentration higher than that of the control. Wines made from grapes treated with BTH and TM, only showed higher concentrations with respect to the control in malvidin-3-O-glucosides and malvidin coumaryglucosides (cis- and trans-). Wines obtained from grapes treated with MeJ + BTH, barely increased some of their principal anthocyanins, not enough to produce significant differences from the control wines. In the same year, 2016, the oenological technique PFCM + TM slightly increased the difference between the control wine and those made from grapes treated with elicitors, especially as regards TriOH anthocyanin compounds. Wines made from grapes treated with BTH were the only ones that presented higher concentrations of total anthocyanins than the control. It should be noted that this cold pre-fermentation increased the extraction of some minority anthocyanin, practically doubling the concentration of vitisins, peonidin acetate, and petunidin coumarate, and even increasing (x4) the concentration of delphinidin acetate.

The wines in 2017 made from grapes treated with MeJ and MeJ + BTH, and elaborated with TM (Table 5), presented lower concentrations of anthocyanins than the control, especially as regards malvidin-3-O-glucoside, and most acetates and coumarates. Wines elaborated with grapes treated with BTH alone did not differ from control wines. In the same year, PFCM + TM led to the increased extraction of most anthocyanins from BTH-treated grapes, as in 2016; non-acylates anthocyanins, acylates, DiOH and TriOH, acetates and coumarates were increased; in fact, it was the only wine that improved its concentration of anthocyanins with respect to the control wine. The wines elaborated from grapes treated with MeJ + BTH presented a higher concentration of delphinidin, cyanidin, and peonidin 3-O-glucoside, surpassing the control wine in DiOH compounds. Other authors also found an increase in anthocyanins in Tempranillo, Merlot, and Syrah wines when pre-fermentative cold maceration was used [48,49].

#### 3.2.3. HPLC analysis of Flavonols

The wines produced by TM in 2016 using grapes treated with elicitors (Table 6) did not present differences in the concentration of flavonols with respect to the control wines (which was not considered surprising in light of the fact that the treatments did not produce significant increases in the same grapes). In this case, the highest flavonol concentrations were obtained for quercetin-3-galactoside and laricitrine-3-glucoside. However, the oenological technique PFCM + TM increased the extraction of flavonols over control levels, particularly in wines made with grapes treated with MeJ and BTH. The impact of using the PFCM + TM technique for winemaking is of interest since it led to a substantial decrease in the concentration of quercetin-3-galactoside and a substantial increase in the extraction of quercetin-3-glucuronide, myricetin, quercetin and laricitrin-3-glucoside.

In the 2017 vinifications using TM (Table 7), wines made from grapes treated with MeJ and MeJ + BTH presented lower concentrations of flavonols than the control wines, while higher concentrations of quercetin, kampferol and siringetin were detected in wines elaborated with grapes treated with BTH. On the other hand, the results of using PFCM + TM hardly changed from those obtained by TM. There was a slight increase in flavonol extraction in wines made from BTH-treated grapes, which were the only wines with a higher concentration of total flavonols compared to the control wines.

As occurred in the comparison of anthocyanin profiles in grapes and wines, differences were found between the grape and wine profiles of flavonols, the grapes were produced using different agricultural techniques (the application of elicitors in our case) and react differently, depending on the climatic conditions, while the different processes of maceration may also have influenced the concentration of flavonols analyzed. It is therefore difficult to talk of a characteristic flavonols profile in Monastrell, such as that detected by Makris et al. [45] and Martínez-Pinilla et al. [50], who mentioned quercetin-3-glucuronide as one of the major flavonols in Monastrell wines.

### 3.3. Multivariate Discriminant Analysis

To improve the visualization and interpretation of all the results obtained, and to check whether the measured variables could distinguish between pre-established groups (vine treatments with elicitors and types of maceration), a multivariate discriminant analysis was made. This analysis revealed that, independently of the meteorological conditions during both campaigns, treatment with elicitors increased the concentrations of phenolic compounds in the treated grapes (Figure 4A); achieving a clear distinction among treatments. However, this distinction was higher in 2016 than in 2017, and the treatment with MeJ produced the greatest increases. Seven discriminant functions were obtained, the first explaining 90% of the variance. The standardized coefficients of this discriminant function showed that the group of acetates and coumarates in anthocyanins, and quercetin-3-glucoside and laricitrin-3-glocoside in flavonols contributed to the most discrimination between treatments.

The discriminant analysis between the two types of maceration used, allowed to distinguish between traditional maceration and that in which cold was applied, especially in 2016 (Figure 4B); showing a higher proportion of phenolic compounds by the application of low temperatures. However, during 2017 an overlap between the results was observed. This lower distinction between the groups analyzed, could be explained by the lower increase of phenolic compounds reached in the grapes during 2017. In this case, three discriminant functions were obtained in wines; the first function explained 92% of the variance and the variables with the greatest discriminant power were color intensity, total anthocyanins and total phenols.

## 4. Conclusions

The higher precipitation, together with a greater number of days with temperatures above 30 °C between July and August 2017, induced variations in the processes of Monastrell maturation, affecting to a greater extent the response to the treatments in the vineyard, compared with 2016. These changes included an earlier beginning to veraison, increases in berry size (generating the dilution of phenolic compounds), the degradation or inhibition of anthocyanin synthesis, as well as an increase in some minor flavonols. Spectrophotometric measurements in wine samples showed that the application of the PFCM + TM technique allowed the phenolic compounds to be extracted to a greater extent in those grapes that increased their phenolic composition through the use of elicitors, giving wines with a higher index of total phenols, a greater intensity of color, and a lower luminosity. Finally, the discriminant analysis revealed that independently of the meteorological conditions during both years, the different treatments with elicitors induced increases in the concentrations of phenolic compounds in the treated grapes, allowing the distinction of the treatments. This analysis allowed the distinction between the types of maceration used, showing a higher proportion of phenolic compounds by the application of low temperature.

## Figures and Tables

**Figure 1 biomolecules-09-00671-f001:**
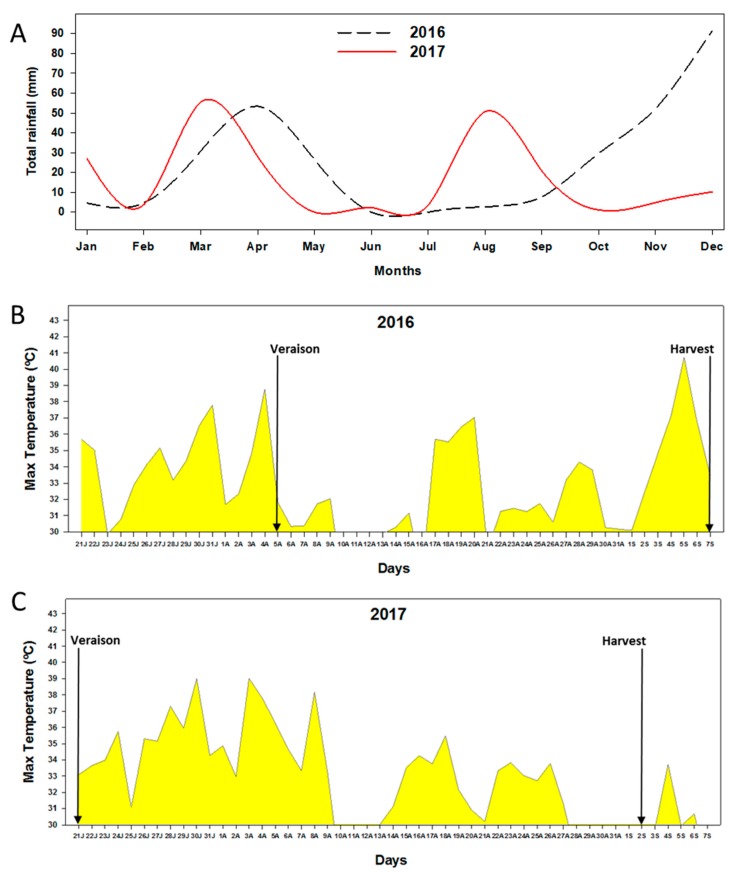
(**A**) Monthly evolution of rainfall recorded on the experimental farm during 2016 and 2017. (**B**) Daily evolution of maximum temperatures (above 30 °C) recorded on the experimental farm during 2016. (**C**) Daily evolution of maximum temperatures (above 30 °C) recorded on the experimental farm during 2017.

**Figure 2 biomolecules-09-00671-f002:**
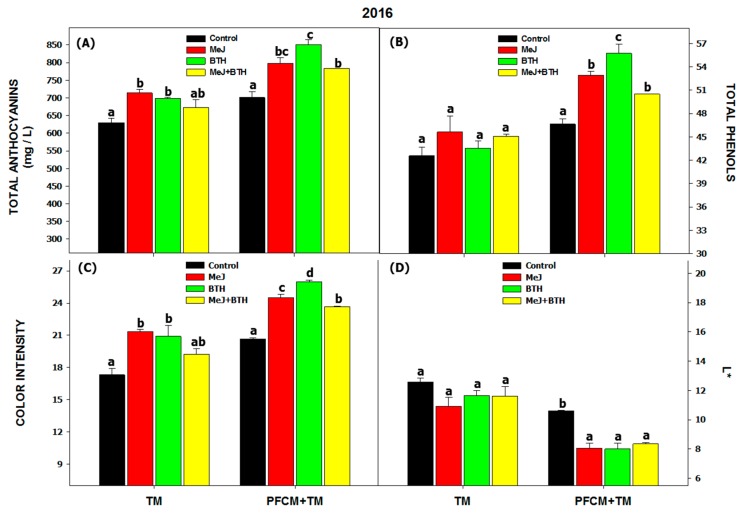
Wine chromatic characteristics in 2016. Total Anthocyanins (**A**), Total Phenols (**B**), Color intensity (**C**), and Luminosity (L*) (**D**). Different letters within the same oenological technique (TM or PFCM + TM) indicate significant differences according to the Duncan test (*p* < 0.05). Abbreviations: MEJ: methyl jasmonate; BTH: benzothiadiazole; TM: traditional maceration; PFCM: pre-fermentative cold maceration.

**Figure 3 biomolecules-09-00671-f003:**
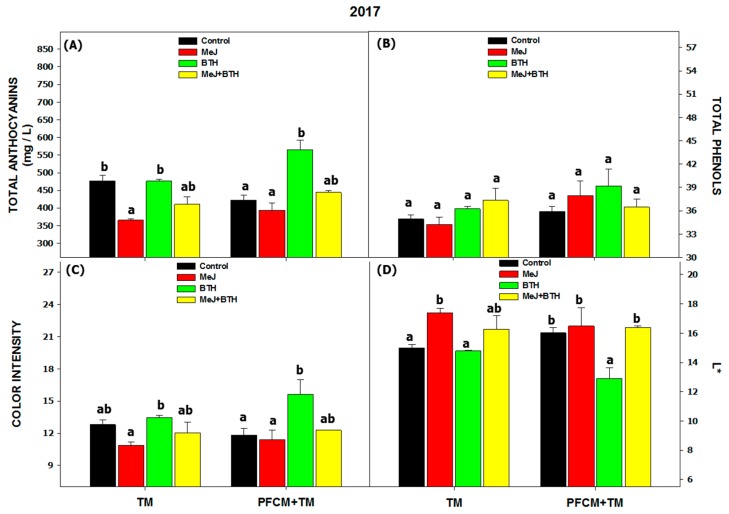
Wine chromatic characteristics in 2017. Total Anthocyanins (**A**), Total Phenols (**B**), Color intensity (**C**), and Luminosity (L*) (**D**). Different letters within the same oenological technique (TM or PFCM + TM) indicate significant differences according to the Duncan test (*p* < 0.05). Abbreviations: MEJ: methyl jasmonate; BTH: benzothiadiazole; TM: traditional maceration; PFCM: pre-fermentative cold maceration.

**Figure 4 biomolecules-09-00671-f004:**
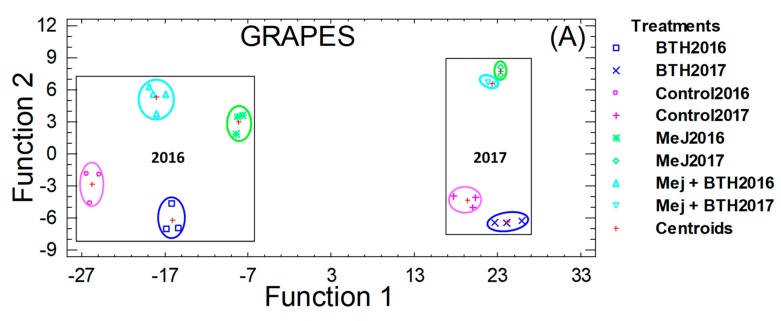
Distribution of the grapes (**A**) and wines (**B**) samples, in the coordinate system defined by the discriminants functions, used to differentiate among treatments.

**Table 1 biomolecules-09-00671-t001:** Concentration of anthocyanins by HPLC in Monastrell grape berry skin treated with MeJ, BTH and MeJ + BTH.

Anthocyanins (*µg/g skin*)	2016				2017			
	Control	MeJ	BTH	MeJ + BTH	Control	MeJ	BTH	MeJ + BTH
Del	1129 ± 161 a	1457 ±± 255 b	1361 ± 34 ab	1401 ± 57 b	759 ± 21 a	859 ± 120 a	783 ± 56 a	1039 ± 49 b
Cyan	1271 ± 6 a	1442 ± 19 b	1296 ± 40 a	1708 ± 21 c	511 ± 42 a	1145 ± 270 b	573 ± 124 a	1120 ± 70 b
Pet	1276 ± 81 a	1667 ± 117 c	1609 ± 16 c	1492 ± 8 b	889 ± 17 a	846 ± 120 a	885 ± 55 a	1061 ± 43 b
Pn	1387 ± 102 a	1731 ± 6 b	1673 ± 24 ab	1684 ± 17 ab	567 ± 52 a	749 ± 137 b	539 ± 43 a	995 ± 1 c
Malv	4108 ± 195 a	5579 ± 587 b	5714 ± 171 b	4535 ± 367 a	3048 ± 187 b	2080 ± 246 a	2675 ± 295 b	2968 ± 106 b
Del Ac	31 ± 9 a	46 ± 4 b	42 ± 4 ab	44 ± 3 ab	29 ± 2 a	30 ± 2 a	31 ± 1 a	32 ± 2 a
Cyan Ac	30 ± 3 a	44 ± 3 bc	39 ± 4 b	45 ± 3 c	22 ± 1 a	33 ± 4 b	23 ± 1 a	35 ± 1 b
Pet Ac	54 ± 7 a	90 ± 9 c	85 ± 8 bc	75 ± 4 b	53 ± 2 ab	46 ± 8 a	52 ± 3 ab	58 ± 1 b
Pn Ac	41 ± 5 a	59 ± 4 b	58 ± 9 b	52 ± 9 ab	24 ± 1 a	24 ± 3 a	23 ± 1 a	32 ± 0 b
Malv Ac	208 ± 22 a	311 ± 18 c	315 ± 13 c	260 ± 20 b	217 ± 15 c	133 ± 14 a	176 ± 21 b	173 ± 2 b
Malv Coum cis	27 ± 5 a	33 ± 3 a	41 ± 3 b	30 ± 3 a	34 ± 2 c	23 ± 2 a	31 ± 4 bc	25 ± 1 ab
Del Coum	113 ± 23 a	133 ± 9 ab	161 ± 3 c	138 ± 8 b	113 ± 2 b	82 ± 14 a	105 ± 10 b	99 ± 9 ab
Malv Caf	26 ± 0 a	28 ± 4 ab	32 ± 2 b	26 ± 1 a	26 ± 6 a	18 ± 5 a	19 ± 1 a	17 ± 2 a
Cyan Coum	141 ± 2 a	149 ± 10 ab	164 ± 9 b	174 ± 9 b	89 ± 10 a	93 ± 8 ab	86 ± 8 a	113 ± 9 b
Pet Coum	196 ± 31 a	224 ± 23 a	291 ± 4 b	228 ± 15 a	199 ± 6 c	113 ± 25 a	175 ± 24 bc	141 ± 9 ab
Pn Coum	163 ± 17 a	160 ± 41 a	211 ± 34 a	180 ± 26 a	94 ± 8 b	68 ± 7 a	83 ± 10 ab	102 ± 4 b
Mal Coum trans	808 ± 70 a	1105 ± 125 b	1251 ± 39 b	932 ± 92 a	769 ± 58 c	378 ± 50 a	627 ± 117 bc	513 ± 26 ab
**Total anthocyanins **	**11009 ± 524 a**	**14257 ± 1078 b**	**14344 ± 790 b**	**13003 ± 842 b**	**7441 ± 258 b**	**6721 ± 178 a**	**6885 ± 442 ab**	**8522 ± 73 c**
**Total nonacylated**	**9171 ± 350 a**	**11876 ± 1055 b**	**11654 ± 677 b**	**10820 ± 675 b**	**5774 ± 185 a**	**5679 ± 80 a**	**5454 ± 288 a**	**7183 ± 128 b**
**Total acylated **	**1838 ± 180 a**	**2380 ± 70 b**	**2690 ± 118 c**	**2184 ± 167 b**	**1668 ± 75 c**	**1042 ± 98 a**	**1431 ± 187 ab**	**1339 ± 56 b**
**Total TRIOH **	**6513 ± 491 a**	**8703 ± 1080 b**	**8685 ± 219 b**	**7428 ± 395 a**	**4696 ± 195 b**	**3785 ± 486 a**	**4343 ± 360 ab**	**5068 ± 198 b**
**Total DIOH **	**2658 ± 149 a**	**3173 ± 35 ab**	**2969 ± 475 ab**	**3392 ± 281 b**	**1078 ± 73 a**	**1894 ± 406 b**	**1111 ± 165 a**	**2115 ± 69 b**
**Total acetates **	**364 ± 41 a**	**550 ± 45 c**	**539 ± 24 bc**	**477 ± 31 b**	**345 ± 14 b**	**265 ± 17 a **	**305 ± 24 b**	**330 ± 4 b**
**Total Coumarates **	**1447 ± 139 a**	**1802 ± 63 b**	**2119 ± 98 c**	**1682 ± 137 b**	**1297 ± 63 c**	**758 ± 76 a**	**1107 ± 163 bc**	**992 ± 58 ab**

Abbreviations: Del, delphinidin 3-O-glucoside; Cyan, cyanidin-3-O-glucoside; Pet, petunidin 3-O-glucoside; Pn, peonidin-3-O-glucoside; Malv, malvidin-3-O-glucoside; Ac, acetylglucosides; Coum, coumarylglucosides; Caf, caffeate glucoside; DiOH. Di-hydroxylated anthocyanins; TriOH. Tri-hydroxylated anthocyanins. Data represent means ± SD. Different letters in the same row and for each year indicate significant differences according to the Duncan test (*p* < 0.05).

**Table 2 biomolecules-09-00671-t002:** Physicochemical characteristics of grapes at harvest.

Year	Treatments	Weight 100 berries	Brix	pH	Total acidity	g/L	
						Tartaric acid	Malic acid
**2016**	Control	148 ± 6 a	26 ± 0 a	3.9 ± 0.0 a	2.9 ± 0.4 a	4.1 ± 0.7 a	1.5 ± 0.2 a
	MeJ	147 ± 8 a	25 ± 0 a	3.8 ± 0.0 a	2.8 ± 0.3 a	4.1 ± 0.8 a	1.3 ± 0.0 a
	BTH	143 ± 11 a	26 ± 1 a	3.8 ± 0.1 a	2.7 ± 0.1 a	3.8 ± 0.4 a	1.2 ± 0.1 a
	MeJ + BTH	163 ± 7 b	26 ± 1 a	3.9 ± 0.0 a	2.7 ± 0.3 a	4.0 ± 0.5 a	1.5 ± 0.2 a

**2017**	Control	160 ± 7 a	23 ± 1 ab	3.9 ± 0.1 b	2.5 ± 0.4 a	3.7 ± 0.5 a	1.2 ± 0.0 ab
	MeJ	157 ± 8 a	22 ± 1 a	3.8 ± 0.1 a	3.1 ± 0.5 a	4.5 ± 1.5 a	1.3 ± 0.1 bc
	BTH	166 ± 10 a	23 ± 1 b	3.9 ± 0.1 b	2.7 ± 0.4 a	3.9 ± 0.6 a	1.1 ± 0.1 a
	MeJ + BTH	167 ± 3 a	22 ± 0 ab	3.7 ± 0.0 a	3.1 ± 0.1 a	4.0 ± 0.7 a	1.3 ± 0.1 c

Data represent means±SD. Different letters in the same column and year indicate significant differences according to the Duncan test (*p* <0.05).

**Table 3 biomolecules-09-00671-t003:** Concentration of flavonols by HPLC in Monastrell grape berry skin treated with MeJ, BTH and MeJ + BTH.

Flavonols (*µg/g skin)*	2016				2017			
	Control	MeJ	BTH	MeJ + BTH	Control	MeJ	BTH	MeJ + BTH
M-3-glc	12.8 ± 2.1 a	16.4 ± 2.6 a	16.3 ± 3.9 a	11.6 ± 1.0 a	10.4 ± 0.8 b	5.1 ± 1.1 a	11.1 ± 0.4 b	6.1 ± 0.9 a
Q-3-glc	26.3 ± 8.0 a	23.0 ± 1.0 a	25.1 ± 3.9 a	20.8 ± 4.5 a	21.0 ± 2.1 b	20.2 ± 2.1 b	23.6 ± 5.1 b	13.7 ± 1.7 a
K-3-gal	0.5 ± 0.1 a	0.5 ± 0.2 a	0.4 ± 0.0 a	0.4 ± 0.2 a	1.5 ± 0.1 a	3.0 ± 0.3 c	2.7 ± 0.7 bc	2.0 ± 0.4 ab
K-3-glc + S-3-glc	2.5 ± 1.1 a	2.4 ± 0.4 a	2.9 ± 0.3 a	2.9 ± 0.9 a	0.1 ± 0.0 a	0.5 ± 0.4 ab	0.9 ± 0.1 b	0.7 ± 0.1 b
I-3-glc	1.0 ± 0.1 a	1.0 ± 0.5 a	1.0 ± 0.6 a	0.9 ± 0.4 a	9.7 ± 0.7 ab	12.0 ± 2.5 b	9.3 ± 0.3 ab	8.1 ± 1.7 a
Q-3glcU	8.1 ± 2.2 a	8.3 ± 1.3 a	8.7 ± 1.4 a	9.1 ± 1.5 a	0.1 ± 0.1 a	0.1 ± 0.0 a	0.4 ± 0.1 b	0.3 ± 0.1 b
**Total flavonols **	**51.2 ± 13.1 a**	**51.6 ± 2.1 a**	**54.4 ± 9.3 a**	**45.7 ± 8.0 a**	**42.8 ± 2.7 b**	**40.8 ± 4.9 b**	**47.9 ± 6.1 b**	**30.9 ± 3.5 a**
**Myricetin**	**12.8 ± 2.1 a**	**16.4 ± 2.6 a**	**16.3 ± 3.9 a**	**11.6 ± 1.0 a**	**10.4 ± 0.8 b**	**5.1 ± 1.1 a**	**11.1 ± 0.4 b**	**6.1 ± 0.9 a**
**Quercetin**	**34.3 ± 9.8 a**	**31.3 ± 1.8 a**	**33.8 ± 5.1 a**	**29.9 ± 5.9 a**	**21.1 ± 2.1 b**	**20.3 ± 2.1 b**	**24.0 ± 5.0 b**	**14.0 ± 1.7 a**
**Kaempferol**	**3.0 ± 1.1 a**	**2.9 ± 0.3 a**	**3.3 ± 0.2 a**	**3.3 ± 1.1 a**	**1.6 ± 0.2 a**	**3.5 ± 0.6 b**	**3.5 ± 0.7 b**	**2.7 ± 0.4 b**
**Isorhamnetin**	**1.0 ± 0.1 a**	**1.0 ± 0.5 a**	**1.0 ± 0.6 a**	**0.9 ± 0.4 a**	**9.7 ± 0.7 ab**	**12.0 ± 2.5 b**	**9.3 ± 0.3 ab**	**8.1 ± 1.7 a**
**Syringetin**	**0.64 ± 0.27 a**	**0.60 ± 0.10 a**	**0.72 ± 0.07 a**	**0.71 ± 0.22 a**	**0.02 ± 0.01 a**	**0.13 ± 0.10 ab**	**0.21 ± 0.02 b**	**0.18 ± 0.03 b**

Abbreviations: M, myricetin; Q, quercetin; K, kaempferol; I, isorhamnetin; glc, O-glucoside; gal, O-galactoside; glcU, O-glucoronide; and S, Syringetin-3-glocoside. Data represent means±SD. Different letters in the same row and for each year indicate significant differences according to the Duncan test (*p* <0.05).

**Table 4 biomolecules-09-00671-t004:** Concentration of anthocyanins expressed as mg/L at the end of alcoholic fermentation in Monastrell wines (**2016**).

Anthocyanins.	Traditional maceration				Pre-fermentative cold maceration + traditional maceration			
	Control	MeJ	BTH	MeJ + BTH	Control	MeJ	BTH	MeJ + BTH
Del	41 ± 5 a	60 ± 1 c	49 ± 5 ab	55 ± 3 bc	46 ± 2 a	54 ± 0 b	58 ± 1 c	60 ± 0 c
Cyan	17 ± 2 a	23 ± 2 a	18 ± 5 a	25 ± 2 a	24 ± 1 ab	21 ± 2 a	22 ± 2 ab	26 ± 0 b
Pet	77 ± 6 a	98 ± 2 b	86 ± 7 ab	87 ± 2 ab	79 ± 1 a	89 ± 1 b	95 ± 4 b	93 ± 4 b
Pn	42 ± 4 a	55 ± 3 ab	46 ± 8 ab	55 ± 1 b	58 ± 1 a	49 ± 7 a	54 ± 7 a	54 ± 7 a
Malv	346 ± 2 a	408 ± 17 b	386 ± 11 b	335 ± 16 a	319 ± 4 a	368 ± 16 bc	403 ± 19 c	353 ± 5 ab
Vitisin A	4.0 ± 0.0 a	3.2 ± 3.1 a	2.6 ± 2.2 a	2.6 ± 2.3 a	7.1 ± 3.0 a	5.3 ± 0.1 a	4.8 ± 0.3 a	5.9 ± 0.5 a
Del acetate	3.4 ± 0.1 a	2.5 ± 1.7 a	2.5 ± 2.0 a	2.7 ± 1.7 a	20 ± 10 a	13 ± 1 a	13 ± 1 a	14 ± 0 a
Vitisin B	3.8 ± 2.4 a	1.8 ± 1.2 a	2.1 ± 1.9 a	1.9 ± 1.1 a	8.8 ± 4.4 a	10.0 ± 3.7 a	8.2 ± 2.3 a	11 ± 3 a
Acetyl vitisin A	6.9 ± 0.6 a	7.0 ± 1.1 a	5.9 ± 2.4 a	7.1 ± 0.8 a	9.4 ± 8.9 a	6.9 ± 4.5 a	6.4 ± 4.2 a	4.6 ± 1.8 a
Cyan acetate	9.1 ± 0.8 a	8.5 ± 2.3 a	8.3 ± 3.3 a	9.3 ± 2.3 a	12 ± 5 a	9.4 ± 0.2 a	9.0 ± 0.6 a	9.6 ± 0.7 a
Pet acetate	11 ± 1 a	11 ± 2 a	10 ± 3a	11 ± 2 a	7.2 ± 5.9 a	6.5 ± 4.1 a	5.1 ± 2.6 a	6.6 ± 4.4 a
Pn acetate	8.6 ± 0.3 a	8.4 ± 1.1 a	7.9 ± 1.7 a	9.2 ± 1.5 a	15 ± 2 a	15 ± 1a	17 ± 2 a	14 ± 1 a
Malv acetate + Del coum	27 ± 1 a	29 ± 3 a	28 ± 1 a	25 ± 2 a	25 ± 6 a	25 ± 1 a	26 ± 1 a	23 ± 1 a
Pn caf	8.0 ± 0.3 a	8.8 ± 0.7 a	9.4 ± 1.1 a	8.0 ± 0.1 a	9.3 ± 4.0 a	7.6 ± 0.4 a	5.9 ± 0.1 a	7.4 ± 0.7 a
Cyan caf+coum	5.1 ± 0.2 a	4.3 ± 0.8 a ±	4.4 ± 0.0 a	4.1 ± 0.0 a	11 ± 4 a	8.4 ± 0.5 a	9.3 ± 0.5 a	8.3 ± 0.9 a
Pet coum	7.9 ± 0.6 a	8.5 ± 0.1 a	8.8 ± 1.2 a	8.7 ± 0.5 a	15 ± 5 a	14 ± 1 a	18 ± 2 a	12 ± 0 a
Malv coum cis	12.5 ± 0.1 ab	13.8 ± 0.9 bc	15.4 ± 1.3 c	11.2 ± 0.2 a	8.8 ± 2.0 b	4.0 ± 0.2 a	3.9 ± 0.9 a	4.2 ± 1.3 a
Pn coum	7.9 ± 0.1 a	8.1 ± 0.4 a	8.4 ± 0.2 a	7.7 ± 0.5 a	23 ± 5 b	10 ± 4 a	11 ± 2 a	7.9 ± 0.4 a
Malv coum trans	51 ± 3 b	55 ± 3 b	63 ± 1 c	41 ± 4 a	42 ± 0 a	50 ± 1 b	65 ± 1 c	38 ± 3 a
**Total anthocyanins**	**690 ± 22 a**	**813 ± 33 b**	**762 ± 58 ab**	**707 ± 13 a**	**755 ± 20 a**	**766 ± 22 a**	**834 ± 17 b**	**744 ± 26 a**
**Total nonacylated**	**523 ± 19 a**	**643 ± 13 b**	**585 ± 36 ab**	**557 ± 18 a**	**526 ± 19 a**	**581 ± 20 ab**	**632 ± 28 b**	**576 ± 35 ab**
**TriOH**	**464 ± 13 a**	**566 ± 17 c**	**521 ± 23 bc**	**477 ± 21 ab**	**444 ± 10 a**	**511 ± 20 b**	**556 ± 12 c**	**496 ± 5 b**
**DiOH**	**59 ± 6 a**	**78 ± 4 a**	**64 ± 13 a**	**80 ± 3 a**	**82 ± 2 a**	**70 ± 9 a**	**76 ± 9 a**	**80 ± 7 a**
**Total acylated**	**152 ± 0 a**	**158 ± 15 a**	**167 ± 16 a**	**138 ± 2 a**	**204 ± 86 a**	**163 ± 3 a**	**183 ± 16 a**	**146 ± 3 a**
**Acetates**	**60 ± 1 a**	**59 ± 10 a**	**57 ± 11 a**	**57 ± 6 a**	**95 ± 22 a**	**69 ± 8 a**	**70 ± 3 a**	**68 ± 9 a**
**Coumarates**	**84 ± 2 b**	**90 ± 4 b**	**100 ± 4 c**	**72 ± 4 a**	**100 ± 32 a**	**86 ± 5 a**	**107 ± 13 a**	**71 ± 3 a**
**Vitisines**	**15 ± 3 a**	**12 ± 5 a**	**11 ± 6 a**	**12 ± 4 a**	**25 ± 16 a**	**22 ± 8 a**	**19 ± 6 a**	**21 ± 5 a**

Abbreviations: Del, delphinidin 3-O-glucoside; Cyan, cyanidin-3-O-glucoside; Pet, petunidin 3-O-glucoside; Pn, peonidin-3-O-glucoside; Malv, malvidin-3-O-glucoside; Caf, caffeate; Coum, coumarylglucosides; DiOH, di-hydroxylated anthocyanins; TriOH, tri-hydroxylated anthocyanins. Data represent means±SD. Different letters in the same row and for each maceration treatment indicate significant differences according to the Duncan test (*p* < 0.05).

**Table 5 biomolecules-09-00671-t005:** Concentration of anthocyanins expressed as mg/L at the end of alcoholic fermentation in Monastrell wines. (**2017**).

Anthocyanins	Traditional maceration				Pre-fermentative cold maceration + traditional maceration			
	Control	MeJ	BTH	MeJ + BTH	Control	MeJ	BTH	MeJ + BTH
Del	28 ± 3 a	25 ± 1 a	29 ± 0 a	27 ± 5 a	22 ± 2 a	26 ± 1 ab	37 ± 3 c	31 ± 0 b
Cyan	12 ± 1 a	15 ± 3 a	13 ± 1 a	18 ± 3 a	9.9 ± 0.1 a	16 ± 2 b	17 ± 1 bc	19 ± 1 c
Pet	51 ± 4 a	40 ± 1 a	52 ± 1 a	43 ± 8 a	41 ± 4 a	41 ± 5 a	61 ± 10 b	46 ± 0 ab
Pn	31 ± 4 a	26 ± 4 a	28 ± 0 a	29 ± 5 a	25 ± 1 a	28 ± 5 ab	42 ± 3 c	33 ± 1 b
Malv	250 ± 15 b	166 ± 1 a	248 ± 17 b	182 ± 27 a	203 ± 18 a	164 ± 15 a	264 ± 36 b	179 ± 3 a
Vitisin A	3.6 ± 0.7 a	3.0 ± 0.1 a	4.0 ± 0.0 a	4.1 ± 0.4 a	3.8 ± 1.0 a	3.6 ± 0.5 a	4.7 ± 0.3 a	4.1 ± 0.3 a
Del acetate	8.6 ± 0.0 d	6.2 ± 0.1 b	7.5 ± 0.1 c	5.4 ± 0.3 a	6.2 ± 0.1 a	6.2 ± 1.0 a	8.7 ± 0.9 b	6.9 ± 0.1 ab
Vitisin B	2.1 ± 0.2 a	1.7 ± 0.0 a	2.0 ± 0.3 a	2.0 ± 0.0 a	1.9 ± 0.2 a	2.0 ± 0.1 a	2.2 ± 0.2 a	2.2 ± 0.0 a
Acetyl vitisin A	4.6 ± 0.1 a	4.6 ± 0.5 a	5.6 ± 0.7 a	4.8 ± 0.7 a	4.5 ± 0.3 a	4.4 ± 0.5 a	5.0 ± 0.1 a	4.9 ± 0.0 a
Cyan acetate	5.4 ± 0.3 ab	4.0 ± 0.3 a	8.2 ± 0.3 b	5.9 ± 2.0 ab	4.1 ± 0.5 a	5.4 ± 1.8 ab	9.8 ± 0.9 c	8.3 ± 0.1 bc
Pet acetate	6.3 ± 0.3 b	4.6 ± 0.1 a	6.1 ± 0.0 b	4.8 ± 0.4 a	5.3 ± 0.4 a	4.9 ± 0.4 a	6.9 ± 0.0 b	5.3 ± 0.0 a
Pn acetate	9.9 ± 0.7 b	6.5 ± 0.2 a	9.5 ± 0.1 b	6.7 ± 0.5 a	8.0 ± 0.8 a	6.8 ± 1.0 a	11 ± 1 b	7.7 ± 0.2 a
Malv acetate + Del coum	19 ± 1 b	11 ± 0 a	17 ± 1 b	12 ± 1 a	15 ± 1 ab	12 ± 3 a	18 ± 2 b	12 ± 0 a
Pn caf	4.9 ± 0.1 b	2.5 ± 0.1 a	4.6 ± 0.2 b	2.5 ± 0.3 a	3.8 ± 0.1 b	2.7 ± 0.1 a	4.5 ± 0.3 c	3.0 ± 0.1 a
Cyan caf+coum	4.9 ± 0.4 b	3.7 ± 0.0 a	4.6 ± 0.1 b	3.5 ± 0.3 a	4.1 ± 0.0 a	3.8 ± 0.4 a	6.3 ± .0.6 b	4.5 ± 0.1 a
Pet coum	8.0 ± 0.8 b	4.6 ± 0.4 a	8.3 ± 0.3 b	4.7 ± 0.5 a	6.8 ± 0.7 b	4.6 ± 0.2 a	9.3 ± 1.3 c	5.1 ± 0.2 ab
Malv coum cis	3.0 ± 0.2 a	2.0 ± 0.4 a	3.3 ± 0.8 a	2.1 ± 0.1 a	2.9 ± 0.2 b	2.2 ± 0.0 a	3.1 ± 0.3 b	2.1 ± 0.3 a
Pn coum	5.8 ± 0.7 b	3.1 ± 0.1 a	5.0 ± 0.2 b	3.0 ± 0.3 a	4.2 ± 0.0 b	3.0 ± 0.4 a	7.1 ± 0.6 c	3.6 ± 0.1 ab
Malv coum trans	31 ± 3 b	16 ± 1 a	31 ± 1 b	17 ± 2 a	25 ± 3 b	15 ± 1 a	32 ± 4 c	16 ± 1 a
**Total anthocyanins**	**489 ± 32 b**	**346 ± 6 a**	**488 ± 17 b**	**376 ± 56 a**	**397 ± 36 a**	**352 ± 36 a**	**550 ± 71 b**	**395 ± 6 a**
**Total nonacylated**	**372 ± 27 b**	**273 ± 8 a**	**371 ± 17 b**	**298 ± 48 ab**	**301 ± 27 a**	**275 ± 30 a**	**421 ± 58 b**	**309 ± 5 a**
**TriOH**	**328 ± 22 b**	**231 ± 2 a**	**330 ± 19 b**	**251 ± 40 a**	**266 ± 26 a**	**231 ± 24 a**	**362 ± 54 b**	**256 ± 4 a**
**DiOH**	**43 ± 5 a**	**41 ± 6 a**	**41 ± 2 a**	**47 ± 8 a**	**35 ± 1 a**	**44 ± 7 ab**	**59 ± 4 c**	**53 ± 2 bc**
**Total acylated**	**107 ± 5 b**	**64 ± 2a**	**105 ± 1 b**	**67 ± 7 a**	**85 ± 7 a**	**67 ± 5 a**	**117 ± 13 b**	**75 ± 1 a**
**Acetates**	**49 ± 0 b**	**32 ± 0 a**	**49 ± 1 b ± **	**35 ± 4 a**	**38 ± 3 a**	**36 ± 4 a**	**55 ± 6 b**	**40 ± 0 a**
**Coumarates**	**52 ± 5 b**	**29 ± 2 a**	**52 ± 0 b**	**30 ± 3 a**	**43 ± 3 b**	**28 ± 1 a**	**58 ± 6 c**	**32 ± 1 a**
**Vitisines**	**10 ± 1 a**	**9.2 ± 0.4 a**	**12 ± 1 a**	**11 ± 1 a**	**10 ± 1 a **	**10 ± 1 a**	**12 ± 1 a**	**11 ± 0 a**

Abbreviations: Del, delphinidin 3-O-glucoside; Cyan, cyanidin-3-O-glucoside; Pet, petunidin 3-O-glucoside; Pn, peonidin-3-O-glucoside; Malv, malvidin-3-O-glucoside; Caf, caffeate; Coum, coumarylglucosides; DiOH, Di-hydroxylated anthocyanins; TriOH, Tri-hydroxylated anthocyanins. Data represent means±SD. Different letters in the same row and for each maceration treatment indicate significant differences according to the Duncan test (*p* < 0.05).

**Table 6 biomolecules-09-00671-t006:** Concentration of flavonols expressed as mg/L at the end of alcoholic fermentation in Monastrell wines (**2016**).

Flavonols	Traditional maceration				Pre-fermentative cold maceration + traditional maceration			
	Control	MeJ	BTH	MeJ + BTH	Control	MeJ	BTH	MeJ + BTH
M-3-gal	2.6 ± 0.1 a	2.9 ± 0.2 a	2.7 ± 0.8 a	2.9 ± 0.5 a	4.3 ± 0.3 a	6.0 ± 1.3 ab	7.0 ± 0.7 b	5.7 ± 1.0 ab
M-3-glc	1.2 ± 0.1 a	1.1 ± 0.0 a	1.2 ± 0.1 a	1.2 ± 0.1 a	6.9 ± 0.8 a	6.4 ± 0.4 a	7.5 ± 0.2 a	6.4 ± 0.4 a
Q-3-gal	59 ± 9 a	67 ± 5 a	69 ± 15 a	64 ± 7 a	11 ± 4 a	12 ± 4 a	13 ± 4 a	14 ± 1 a
Q-3- glcU	1.9 ± 0.3 a	2.4 ± 0.1 a	2.5 ± 0.7 a	2.3 ± 0.6 a	20 ± 14 a	25 ± 20 a	25 ± 21 a	23 ± 18 a
Q-3-glc	11 ± 1 a	13 ± 1 a	14 ± 3 a	13 ± 2 a	40 ± 1 a	45 ± 3 ab	49 ± 1 b	43 ± 4 ab
L-3-glc	27 ± 4 a	34 ± 1 a	32 ± 6 a	29 ± 0 a	33 ± 5 a	32 ± 4 a	36 ± 7 a	31 ± 3 a
K-3-gal	2.1 ± 0.1 a	1.8 ± 0.1 a	2.3 ± 0.4 a	1.8 ± 0.3 a	5.7 ± 4.4 a	6.0 ± 4.5 a	6.5 ± 4.6 a	6.0 ± 3.8 a
K-3-glc	0.8 ± 0.1 a	0.9 ± 0.0 a	0.9 ± 0.2 a	0.8 ± 0.1 a	5.0 ± 2.2 a	3.8 ± 0.0 a	4.1 ± 0.0 a	4.1 ± 0.6 a
I-3-glc	8.4 ± 1.8 a	9.0 ± 0.8 a	8.4 ± 1.4 a	7.8 ± 0.8 a	9.0 ± 3.2 a	9.2 ± 3.6 a	9.9 ± 4.5 a	9.0 ± 3.8 a
S-3-glc	1.8 ± 0.5 a	1.8 ± 0.3 a	1.6 ± 0.4 a	1.6 ± 0.2 a	5.5 ± 0.7 a	6.1 ± 0.8 a	6.9 ± 0.2 a	5.7 ± 1.2 a
**Total**	**115 ± 17 a**	**133 ± 6 a**	**133 ± 26 a**	**124 ± 11 a**	**141 ± 5 a**	**152 ± 3 b**	**164 ± 1 c**	**148 ± 3 ab**
**Myricetin**	**3.8 ± 0.0 a**	**4.0 ± 0.2 a**	**3.9 ± 0.8 a**	**4.1 ± 0.4 a**	**11 ± 1 a**	**12 ± 2 a**	**14 ± 2 a**	**12 ± 2 a**
**Quercetin**	**72 ± 11 a**	**82 ± 5 a**	**84 ± 18 a**	**79 ± 10 a**	**71 ± 25a**	**82 ± 28 a**	**87 ± 34 a**	**80 ± 30 a**
**Laricitrina**	**27 ± 4 a**	**34 ± 1 a**	**32 ± 6 a**	**29 ± 0 a**	**33 ± 5 a**	**32 ± 4 a**	**36 ± 7 a**	**31 ± 3 a**
**Kaempferol**	**2.8 ± 0.0 ab**	**2.7 ± 0.2 a **	**3.2 ± 0.2 b**	**2.6 ± 0.2 a**	**11 ± 2 a**	**10 ± 5 a**	**11 ± 5 a**	**10 ± 3 a**
**Isorhamnetin**	**8.4 ± 1.8 a**	**9.0 ± 0.8 a**	**8.4 ± 1.4 a**	**7.8 ± 0.8 a**	**9.0 ± 3.2 a**	**9.2 ± 3.6 a**	**9.9 ± 4.5 a**	**9.0 ± 3.8 a**
**Syringetin**	**1.8 ± 0.5 a**	**1.8 ± 0.3 a**	**1.6 ± 0.4 a**	**1.6 ± 0.2 a**	**5.5 ± 0.7 a**	**6.1 ± 0.8 a**	**6.9 ± 0.2 a**	**5.7 ± 1.2 a**

Abbreviations: M, myricetin; Q, quercetin; K, kaempferol; I, isorhamnetin; glc, O-glucoside; gal, O-galactoside; glcU, O-glucoronide; L, laricitrin 3-glucoside; and S, syringetin 3-glucoside. Data represent means±SD. Different letters in the same row and for each maceration treatment indicate significant differences according to the Duncan test (*p* < 0.05).

**Table 7 biomolecules-09-00671-t007:** Concentration of flavonols expressed as mg/L at the end of alcoholic fermentation in Monastrell wines (**2017**).

Flavonols	Traditional maceration				Pre-fermentative cold maceration + traditional maceration			
	Control	MeJ	BTH	MeJ + BTH	Control	MeJ	BTH	MeJ + BTH
M-3-gal	24 ± 0 b	17 ± 1 a	28 ± 1 b	18 ± 3 a	25 ± 1 b	17 ± 2 a	30 ± 2 c	16 ± 1 a
M-3-glc	9.3 ± 0.3 b	5.7 ± 0.1 a	9.2 ± 0.9 b	7.1 ± 1.0 a	7.3 ± 0.7 a	6.6 ± 0.6 a	9.8 ± 1.1 b	7.0 ± 0.6 a
Q-3-gal	5.7 ± 0.1 a	6.1 ± 1.4 a	7.9 ± 0.2 a	6.7 ± 0.6 a	7.4 ± 1.8 a	7.7 ± 1.3 a	6.9 ± 0.2 a	6.0 ± 0.1 a
Q-3- glcU	22 ± 0 a	24 ± 4 a	26 ± 2 a	26 ± 3 a	25 ± 3 a	28 ± 5 a	24 ± 3 a	23 ± 1 a
Q-3-glc	32 ± 2 a	35 ± 3 ab	43 ± 3 b	35 ± 2 ab	38 ± 4 ab	42 ± 4 ab	45 ± 0 b	33 ± 0 a
L-3-glc	5.1 ± 0.1 b	2.8 ± 0.1 a	5.5 ± 0.3 b	3.5 ± 0.5 a	4.8 ± 0.0 bc	3.9 ± 0.2 ab	5.3 ± 0.6 c	3.4 ± 0.3 a
K-3-gal	0.3 ± 0.0 b	0.2 ± 0.0 a	0.9 ± 0.0 d	0.6 ± 0.1 c	0.3 ± 0.0 a	0.6 ± 0.0 b	0.3 ± 0.0 a	0.6 ± 0.1 b
K-3-glc	0.1 ± 0.1 a	0.1 ± 0.1 a	0.4 ± 0.0 b	0.4 ± 0.0 b	0.1 ± 0.0 a	0.4 ± 0.1 b	0.1 ± 0.1 a	0.3 ± 0.0 b
I-3-glc	3.4 ± 0.0 bc	2.4 ± 0.2 a	3.7 ± 0.3 c	2.9 ± 0.3 ab	3.2 ± 0.2 ab	3.5 ± 0.3 ab	3.7 ± 0.2 b	3.0 ± 0.1 a
S-3-glc	2.8 ± 0.3 b	1.6 ± 0.4 a	3.6 ± 0.3 c	2.1 ± 0.2 ab	2.5 ± 0.0 bc	2.0 ± 0.2 ab	2.8 ± 0.2 c	1.8 ± 0.2 a
**Total**	**106 ± 4 ab**	**95 ± 12 a**	**128 ± 6 b**	**103 ± 14 ab**	**114 ± 5 b**	**112 ± 7 b**	**129 ± 5 c**	**95 ± 1 a**
**Myricetin**	**35 ± 2 b**	**22 ± 0 a**	**37 ± 2 b**	**25 ± 4 a**	**32 ± 0 b**	**24 ± 2 a**	**40 ± 5 c**	**23 ± 2 a**
**Quercetin**	**60 ± 5 a**	**66 ± 5 ab**	**77 ± 2 b**	**68 ± 3 ab**	**71 ± 2 b**	**77 ± 3 c**	**77 ± 2 c**	**62 ± 1 a**
**Laricitrina**	**5.1 ± 0.1 b**	**2.8 ± 0.1 a**	**5.5 ± 0.3 b**	**3.5 ± 0.5 a**	**4.8 ± 0.0 bc**	**3.9 ± 0.2 ab**	**5.3 ± 0.6 c**	**3.4 ± 0.3 a**
**Kaempferol**	**0.5 ± 0.0 a**	**0.3 ± 0.1 a**	**1.3 ± 0.1 c**	**1.0 ± 0.1 b**	**0.4 ± 0.1 a**	**1.0 ± 0.1 b**	**0.4 ± 0.1 a**	**0.9 ± 0.0 b**
**Isorhamnetin**	**3.4 ± 0.0 bc**	**2.4 ± 0.2 a**	**3.7 ± 0.3 c**	**2.9 ± 0.3 ab**	**3.2 ± 0.2 ab**	**3.5 ± 0.3 ab**	**3.7 ± 0.2 b**	**3.0 ± 0.1 a**
**Syringetin**	**2.8 ± 0.3 b**	**1.6 ± 0.4 a**	**3.6 ± 0.3 c**	**2.1 ± 0.2 ab**	**2.5 ± 0.0 bc**	**2.0 ± 0.2 ab**	**2.8 ± 0.2 c**	**1.8 ± 0.2 a**

Abbreviations: M, myricetin; Q, quercetin; K, kaempferol; I, isorhamnetin; glc, O-glucoside; gal, O-galactoside; glcU, O-glucoronide; L, laricitrin 3-glucoside; and S, syringetin 3-glucoside. Data represent means±SD. Different letters in the same row and for each maceration treatment indicate significant differences according to the Duncan test (*p* < 0.05).

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
