# Peer review of "Elicitors and Pre-Fermentative Cold Maceration: Effects on Polyphenol Concentration in Monastrell Grapes and Wines"

_biomolecules, 2019, doi:10.3390/biom9110671_

Round 1

Reviewer 1 Report

Biomolecules-610512

Title: Elicitors and pre-fermentative cold maceration: Tools to improve the anthocyanin concentration in Monastrell grapes and wines

In this work, the Methyl Jasmonate and Benzothiadiazole were used as elicitors in order to improve the concentration of anthocyanins and flavonols. These treatements combined with a cold pre-fermentative maceration seems to improve the phenolic extraction. However, the effect of both treatments seems to be uncertain regarding the grape and wine quality and highly conditioned by vintage.

The improvement on the phenolic quality is difficult to evaluate and sometimes is strongly linked to the sensory analysis and the organoleptic properties and this is not supported by results.

From my point of view this work does not meet the quality standards necessary to be published in Biomolecules. The weakest aspect is the obtained results that are not according to this journal.

Author Response

Reviewer 3

List of changes in the current main document / or a rebuttal

Define all abbreviations such as SE in line 14

Line 14. Corrected

Avoid repetitive words such as in line 36 “… skin is thicker than the skin

Line 36. Paragraph has been rewritten.

” Avoid the use of the first person “we, us, I”

Corrected throughout the manuscript.

More specific information such as the amount of sample used, temperature of the sample, among others for the physicochemical and spectrophotometer methods is missing.

For spectrophotometer, at which spectrum each parameter was measured? Even though a reference was added, it must be more specific at least for the most relevant information about the method. This comment also applies to the HPLC method.

Specific information on the methods has been attached.

“… discriminant analysis was applied to identify the most discriminant variables.” In which software was this analysis performed? Which type of discriminant analysis did you use (linear, quadratic, multiple, etc.)? Methods description must be specific

This analysis was performed in Statgraphics 5.0 Plus, and a Multivariate Discriminant Analysis was applied.

Line 139. The details have been corrected in section 2.6 Statistical analysis.

The parenthesis in lines 184-185 should be as a sentence joined to the previous one to give flow for readers

Line 206. Corrected

Results must be presented in the same order as methods; therefore, it is recommended to move the HPLC method before the physicochemical description as it is presented in results.

The methods have been reordered.

Line 213, it must be “In contrast

Line 235. Corrected

…” Figures 2 and 3, letters of significance are wrong; the letters always start with the sample with highest mean; therefore, “a” must be given to the highest, not the lowest.

Statistical software configuration used in our institution, places by default the letter "a" for the lowest average value.

7 tables and 2 figures of the manuscript have the same configuration.

Do you think it would be necessary to reorder the letters of significance of all results?

Description of results in the paragraph starting in line 375 is wrong, it specifies there is an overlap of results from wines in 2017, but in Figure 4b, the overlap appears in 2016.

Paragraph description was correct, but an error was corrected in the figure 4B.

Conclusions section is too long; it should be written in a concise way and a single paragraph

Conclusions section has been rewritten and reduced.

Reviewer 2 Report

Dear Diego Paladines‐Quezada and other Authors

The work is very interesting, touching scientific aspects and interesting for producers.
The separation of material and methods requires supplementation.
The description of the results to a small extent combines the results from individual tables. Laconic references to the effects of polyphenols in fruit on their content in wine.
The large number of results in the tables makes their analysis difficult, there is no statistical analysis comparing maceration methods or individual years.
It would be necessary to insert a figure or a table summarizing the content of polyphenolic compounds - their sum.
There is generally no comparative / statistical analysis between years of research and between fermentation methods. such an analysis would have to be carried out at least for groups of polyphenolic compounds. There is often a statement that there were more polyphenols, but this does not result from statistical analysis.
Especially that the results in individual years are very diverse.
Detailed comments can be found in the text.

Jasmonic acid and its methyl ester also inhibit processes related to plant growth, while stimulating fruit ripening, aging and leaf fall. Has the effect of these compounds been observed on plants. On the basis of many years of research carried out on another species, I found a significant weakening of plant growth and their worse yielding. Would you recommend elicitors for annual use or only for occasional use?

Author Response

Reviewer 2

List of changes in the current main document / or a rebuttal

The work is very interesting, touching scientific aspects and interesting for producers.

The separation of material and methods requires supplementation.

The description of the results to a small extent combines the results from individual tables. Laconic references to the effects of polyphenols in fruit on their content in wine.

The large number of results in the tables makes their analysis difficult, there is no statistical analysis comparing maceration methods or individual years.

It would be necessary to insert a figure or a table summarizing the content of polyphenolic compounds - their sum.

There is generally no comparative / statistical analysis between years of research and between fermentation methods. such an analysis would have to be carried out at least for groups of polyphenolic compounds. There is often a statement that there were more polyphenols, but this does not result from statistical analysis.

Especially that the results in individual years are very diverse.

Detailed comments can be found in the text.

Jasmonic acid and its methyl ester also inhibit processes related to plant growth, while stimulating fruit ripening, aging and leaf fall. Has the effect of these compounds been observed on plants. On the basis of many years of research carried out on another species, I found a significant weakening of plant growth and their worse yielding. Would you recommend elicitors for annual use or only for occasional use?

The supplementation of materials and methods has been carried out, thanks to their observations and that of other reviewers.

The tables and figures of results were described individually, and subsequently a statistical analysis (Multivariate Discriminant Analysis) was performed to summarize and better visualize the results obtained. In this analysis, the effect of elicitor treatments on grapes during two years was compared. As well as the effect of the two types of maceration on the wines obtained.

Thank you very much for the comments based on your experience with Metil Jasmonate. Unfortunately we have no experience of prolonged use for several years in the same vineyard, nor have we analyzed its effect on aging, weakening or falling leaves.

For this reason, at this point in our investigation, we could not recommend its annual use, since we must study its long-term effect.

Detailed corrections are described below.

Comments in PDF

Line 3. studies also concern other parameters, and the title is limited to anthocyanins only

The title has been changed.

Line 76. how many grapes were on the plant, to what level was the yield (t / ha) reduced?

Production results (data not shown) did not show significant differences. Its average value was 3.2 kg/vineyard.

Line 77. the publication contains the same information, incorrect indication

Line 77. Reference corrected.

Line 78. 2, the third is a mixture of both

Line 78. Corrected.

Line 83. were all the plants sprayed with this dose or just control?

Line 82. Corrected. All plants were sprayed with 200 mL.

Line 92. no reference

Line 92. Corrected.

Line 95. dose and what was used?

Line 95-96. A commercial yeast Uvaferm VRB was used (25 g / hL)

Line 102. was the procedure similar to Traditional maceration here?

Line 93-107. In section ¨2.3 Vinification¨ the types of maceration are explained in detail. The results tables are detailed in separate columns, those wines only with Traditional Maceration, and wines with Pre-fermentative cold maceration + Traditional maceration.

Line 103-104. cap was punched down twice a day?

Line 103-104. It is a common work in the winary during the maceration process, to remove the floating pomace (cap) and mix it with the must.

Line 117. in the indicated publication, the authors cite;

Bautista-Ortín A.B., Martínez-Cutillas A., Ros-García J.M., López- Roca J.M. and Gómez-Plaza E., 2005. Improving colour extraction and stability in red wines: the use of maceration enzymes and enological tannins. Int. J. Food Sci. Technol., 40, 867-878.

Line 112. Reference corrected.

Line 131. which year was typical for a given region? results should be included showing the temperature and precipitation for many years.

It is difficult to estimate how many years are needed to talk about a typical year of a region. For this reason, those paragraphs that referred to "a typical year" have been deleted.

Line 133. Table 1.  in berries or skin

Line 166. Table 1. In skin……Corrected

Line 154-155. Figure 1B. what does this horizontal line mean?

Line 188-196. Figure 1B and 1C. This line has been deleted.

Line 179. What was the sum of precipitation during this period?

Line 201. Updated information

Line 195. what were the differences in the days? how long was the beginning of veraison until harvestin

Line 149-151. The information has been added to the manuscript.

Line 203. correct the title of tables 1 and 3, the values do not refer to grape berries but to skin

Line 166-168 (Table 1); and Line 244-246 (Table 3) Titles of these tables have been changed.

Line 213. could also result from a smaller fruit weight. anthocyanins are mainly in the skin, and in the same weight small fruits have a larger skin surface than larger fruits

This could also be an explanation, thanks for the suggestion, but we have no data on skin diameter or surface, we cannot relate it to weight.

Line 224-225. Table 3.

it would be advisable to use the same units - Tables 1,3 and 4-7

The units in Tables 1 and 3 are the same (grape analysis). The units in tables 4-7 are different, since they represent wine analysis

Line 229. what level? how much more than normal?

Line 250. Paragraph has been rewritten

Line 267. so it was darker

Line 288. Yes, it was. It is a description based on the color parameters.

Line 272-273. Figure 2. it is difficult to analyze the results if they are presented in different charts.

It was advisable to place the examined feature next to each other in individual years

Line 293-297. Figure 2. The results are presented in years separately. We think that this way of presenting the results, would not cause difficulty in interpretation.

´Line 286. very general statement. no comparative analysis of these tables - fruit / skin: wine

Have you tested the must / wine parameters after cold maceration? what was the degree of extraction of Anthocyanins after this period?

The results have been presented describing the tables or graphs individually. A Multivariate Discriminant Analysis is presented below, in which groups of results are compared.

In this study the degree of anthocyanin extraction was not verified, since previous studies have shown that cold pre-fermentation maceration allows a greater proportion of phenolic compounds to be extracted (Referenced in ¨Introduction¨)

Line 390. Conclusions

In general, since the years of the research had very varied weather, one cannot draw conclusions based on the results from 2016, completely excluding 2017.

Since 2016 was typical and 2017 was not, then it would be necessary to repeat the research for another year to verify the results obtained.

Line 412. The "Conclusions" section has been rectified.

Line 400-402

no information on typical conditions for this region.

The only thing that can be concluded is that in both years the use of BTH gave a positive effect.

based on research from 2016, one should not write such definitive conclusions: we can conclude that the pre ‐ harvest response to the application of elicitors MeJ, BTH, and MeJ + BTH, increased the total anthocyanins concentration analyzed by HPLC, including DiOH, TriOH , acetates and coumarates present in Monastrell grapes

As mentioned above. It is difficult to estimate how many years are needed to talk about a typical year of a region. For this reason, those paragraphs that referred to "a typical year" have been deleted.

Line 412. The "Conclusions" section has been rectified.

Reviewer 3 Report

The study is interesting; however, the manuscript needs some improvements:

Define all abbreviations such as SE in line 14 Avoid repetitive words such as in line 36 “… skin is thicker than the skin …” Avoid the use of the first person “we, us, I” More specific information such as the amount of sample used, temperature of the sample, among others for the physicochemical and spectrophotometer methods is missing. For spectrophotometer, at which spectrum each parameter was measured? Even though a reference was added, it must be more specific at least for the most relevant information about the method. This comment also applies to the HPLC method. “… discriminant analysis was applied to identify the most discriminant variables.” In which software was this analysis performed? Which type of discriminant analysis did you use (linear, quadratic, multiple, etc.)? Methods description must be specific The parenthesis in lines 184-185 should be as a sentence joined to the previous one to give flow for readers Results must be presented in the same order as methods; therefore, it is recommended to move the HPLC method before the physicochemical description as it is presented in results. Line 213, it must be “In contrast …” Figures 2 and 3, letters of significance are wrong; the letters always start with the sample with highest mean; therefore, “a” must be given to the highest, not the lowest. Description of results in the paragraph starting in line 375 is wrong, it specifies there is an overlap of results from wines in 2017, but in Figure 4b, the overlap appears in 2016. Conclusions section is too long; it should be written in a concise way and a single paragraph

Author Response

(The authors gave the same response as above.)

Round 2

Reviewer 1 Report

The title of the article has been changed, this work is now more concise and lower ambitious. Changes suggested by reviewers have been considered in the new version of the manuscript and some paragraph has been rectified.